# Exploring the Potential of Carbon Capture, Utilization, and Storage in Baltic Sea Region Countries: A Review of CCUS Patents from 2000 to 2022

**Mayur Pal \***, **Viltė Karaliūtė and Shruti Malik**

Department of Mathematics, Kaunas University of Technology, 44249 Kaunas, Lithuania
* Correspondence: mayur.pal@ktu.lt

**Abstract:** Carbon capture, utilization, and storage (CCUS) refers to technologies that capture carbon dioxide ($CO_2$) emissions from sources such as power plants, industrial facilities, and transportation, and either store it underground or use it for beneficial purposes. CCUS can play a role in reducing greenhouse gas emissions and mitigating climate change, as $CO_2$ is a major contributor to global warming. In the Baltic Sea region countries (BSR), patent searches from 2000 to 2020 reveal that CCUS technologies are focused on $CO_2$ storage, monitoring, utilization, and transport. However, the adoption and deployment of these technologies has been slow due to a variety of factors, including a lack of government action on climate change, public skepticism, increasing costs, and advances in other options such as renewables and shale gas. Overall, CCUS has the potential to significantly reduce $CO_2$ emissions and contribute to climate change mitigation efforts, but more work is needed to overcome the barriers to its widespread adoption in the BSR and elsewhere. This could include policy measures to incentivize the use of CCUS technologies, public education and outreach efforts to increase understanding and support for CCUS, and research and development to improve the efficiency and cost-effectiveness of these technologies.

**Keywords:** carbon capture; utilization; storage; CCUS; patents; Baltic Sea region countries; carbon reduction





## 1. Introduction

Capturing $CO_2$ has been suggested as a way to mitigate climate change since the 1970s. $CO_2$ capture technologies have also been used for a variety of purposes for decades. One example of this is the separation of $CO_2$ from natural gas reservoirs. This process, known as gas sweetening, is used to remove impurities such as $CO_2$ and hydrogen sulfide from natural gas so that it can be used as a relatively cleaner burning fuel.

$CO_2$ capture technologies can be divided into two main categories: pre-combustion capture and post-combustion capture. Pre-combustion capture involves separating $CO_2$ from a fuel before it is burned, while post-combustion capture involves capturing $CO_2$ from the flue gases produced by the combustion of a fuel [1].

There are a variety of different technologies that can be used for $CO_2$ capture, including chemical absorption, membrane separation, and cryogenic separation. These technologies have been used in a variety of applications, including power generation, cement production, and the production of chemicals and fuels [2].

Enhanced oil recovery (EOR) using $CO_2$ injection has been used for many years to increase the amount of oil that can be recovered from a reservoir. In this process, $CO_2$ is injected into an oil reservoir to reduce the viscosity of the oil, thus making its recovery easier. The $CO_2$ can be captured from a variety of sources, including natural underground reservoirs and industrial facilities. EOR using $CO_2$ injection can be a cost-effective way to increase oil recovery and reduce greenhouse gas emissions, as the $CO_2$ that is injected into

the reservoir is typically not released into the atmosphere. Instead, it is permanently stored in the reservoir [3].

The Sleipner project in Norway [4], which was the world's first integrated carbon capture and storage (CCS) project and began operating in 1996, captures $CO_2$ from a natural gas processing facility and injects it into a deep saline aquifer in the North Sea for permanent storage. The Sleipner project has demonstrated the feasibility of large-scale CCS and has been a valuable learning opportunity for the development of CCS technologies.

Carbon capture and storage (CCS) has gained significant attention in recent years as a potential tool for reducing greenhouse gas emissions. CCS involves capturing $CO_2$ from industrial sources, such as power plants and factories, compressing it, and transporting it to a storage site where it can be injected underground for permanent storage. CCS is seen as a particularly promising technology for certain industries that are difficult to decarbonize, such as cement, iron and steel, and chemical production, which tend to have high emissions of $CO_2$.

CCS is expected to play a significant role in meeting the emissions reduction targets set out in the Paris Agreement, which aims to limit global warming to well below 2 °C as compared to pre-industrial levels [5]. The Intergovernmental Panel on Climate Change (IPCC) estimates that CCS could contribute up to 19% of the emissions reductions needed by 2050 to meet this goal.

While CCS has the potential to significantly reduce $CO_2$ emissions, it also has some challenges and limitations. For example, it can be energy intensive and costly, and the captured $CO_2$ must be stored in a secure and environmentally safe manner. A risk analysis has been done related to financial, technical, environmental, and health and safety aspects that might affect the implementation of CCUS [6]. Despite these challenges, research and development efforts are ongoing to improve the efficiency and cost-effectiveness of CCS technologies. A multiscale framework has been proposed to design and optimize a cost effective CCUS and CCS supply chain network in the USA, which included screening of materials, processes, and technologies for $CO_2$ capture [7].

Carbon capture and storage (CCS) is a technology that separates carbon dioxide ($CO_2$) from other gases and stores it in the subsurface. The industrial sector, which includes industries such as iron and cement manufacturing, could greatly benefit from this technology, as these industries currently produce large amounts of $CO_2$ as a byproduct of their operations. CCS offers a way to capture and store this $CO_2$ rather than releasing it into the atmosphere [1,2]. Additionally, there are many potential storage sites located near major point sources of $CO_2$, such as power plants and factories (Figure 1). While the CCS industry is still developing, there is room for growth. In 2020, CCS installations had the capacity to capture approximately 38.5 million metric tons of $CO_2$ per year, which is a small number compared to the global total of $CO_2$ emitted annually. If we don't capture $CO_2$ and remove it from atmosphere, the natural systems that keep Earth's climate relatively peaceful and comfortable shall start to tip. The shift will be chaotic, and the new normal might not be conducive to life as we know it.

Another option is to use renewable or nuclear energy sources to generate electricity for our homes, offices, and vehicles. However, nuclear power is expensive and lacks public support, and renewable energy sources may face challenges in finding enough land for its deployment at a large scale. Additionally, some industries, such as aviation and iron smelting, currently produce $CO_2$ as a byproduct of their operations and may be difficult to decarbonize without additional technologies. This is where carbon capture and storage (CCS) comes into the picture. By capturing $CO_2$ from point sources such as power plant flue gas and storing it underground, we can significantly reduce $CO_2$ emissions. Other technologies, such as direct air capture, can remove $CO_2$ that is already present in the atmosphere. However, CCS and other carbon capture technologies need to become much larger, cheaper, and more efficient in order to make a significant impact on global emissions.

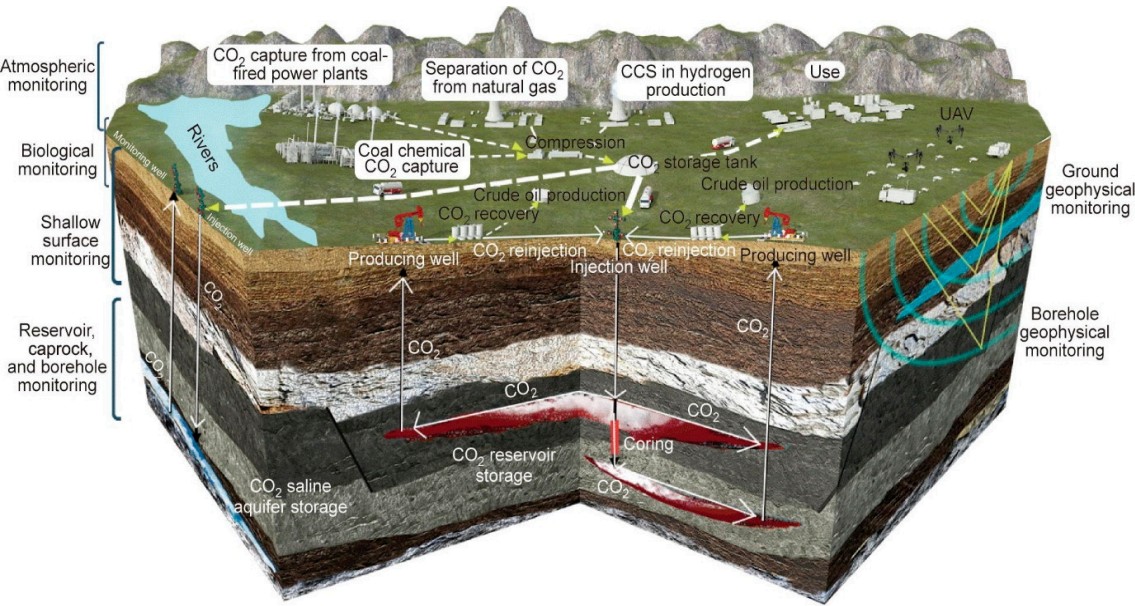

**Figure 1.** The CCS industry chain [2].

The use of machine learning (ML) algorithms to study the different aspects of CCUS is increasing with advancement in technology. It not only reduces the simulation cost but also offers a quick and effective method to perform capture, storage, transportation, and utilization processes. An overview on usage of ML at every step of CCUS has been presented in one study [8]. Another study [9], presented the application of ML depending upon the research objective in CCUS, such as estimation of petrophysical properties or leak detection. Each of these objectives use a different ML algorithm to perform the task. Hybrid models have been also proposed to determine the deliverability of underground natural gas storage sites in USA [10] which can be used to store gases like $CO_2$ or Hydrogen. ML has also been used to predict $CO_2$ trapping efficiency [11] or studying the impact of interfacial tension (IFT) in $CO_2$-brine system on $CO_2$ sequestration [12]. Thus, offering a cheaper and efficient method to achieve large-scale CCS.

European Union Strategy for the Baltic Sea Region (EUSBSR) is a regional strategy that was approved by the European Council in 2009 [13–17]. It is the first of its kind in Europe. The EUSBSR involves cooperation between eight EU member states in the Baltic Sea region: Sweden, Denmark, Estonia, Finland, Germany, Latvia, Lithuania, and Poland. The Strategy also welcomes cooperation with neighboring countries outside of the EU, including Russia, Iceland, Norway, and Belarus. All of these countries, including the neighboring countries not directly within the EUSBSR, are referred as the Baltic Sea region (BSR) countries.

By November 2019, all of the countries in the Baltic Sea region (BSR), including Russia, had ratified the Paris Climate Agreement. By November 2020, most of these countries had set strategic climate targets for 2030 and 2050. One technology that can help these countries to meet their climate targets is carbon capture, utilization, and storage (CCUS). CCUS technology is effective at significantly reducing $CO_2$ emissions and can play a key role in the transition to a low-carbon economy.

The Baltic Basin (BB) is a region with significant potential for storing $CO_2$ in its sedimentary rocks. Countries within the BB border, such as Sweden, Latvia, Lithuania, Poland, and Russia, have identified large saline aquifers and depleted oil and gas fields as particularly promising areas for $CO_2$ storage [16,17]. Especially in Baltic states, the $CO_2$ emission sources and potential storage sites have been explored and identified [18]. In Latvia, the Dobele structure is being explored for utilization as an underground natural gas storage facility, to meet the needs of natural gas in Latvia and other Baltic states [19]. In Lithuania, the attempts to present an initial assessment of the storage potential of $CO_2$ in saline aquifers and depleted oil and gas fields has been carried out [20]. Although

this estimation was based on volumetric assessment, coordinating such research and development efforts across the Baltic Sea region (BSR) is important to fully utilize the potential of CCUS in the region.

Russia has recently ratified the Paris Climate Agreement and is considering the introduction of a carbon tax and emissions trading scheme in 2025. The country has also released a draft long-term strategy for reducing greenhouse gas emissions and diversifying economic development by 2050. However, even under the most ambitious scenario in this strategy, Russia will only reach carbon neutrality "close to the end" of the century. The Russian government has also approved a 4-year plan for the development of hydrogen energy, which will support pilot projects for the production of hydrogen without $CO_2$ emissions [21–30]. Overall, it is clear that while Russia is taking steps to address climate change and reduce $CO_2$ emissions, more needs to be done to accelerate the adoption of CCUS technologies and achieve the necessary emissions reductions to meet global climate goals.

The process of removing carbon dioxide ($CO_2$) from the Earth's atmosphere involves several steps and technologies, including:

1. Site selection: Identifying a suitable location for carbon capture and storage (CCS) is an important first step. This may involve evaluating factors such as the geology of the area, the proximity to point sources of $CO_2$, and the availability of infrastructure for transportation and storage.
2. Capture technology: There are a variety of technologies that can be used to capture $CO_2$ from the atmosphere or from point sources such as power plants and factories. These technologies include chemical absorption, membrane separation, and cryogenic separation.
3. Transport: Once $CO_2$ is captured, it must be transported to a storage site. This can be done by truck, train, or pipeline, depending on the location of the storage site and the volume of $CO_2$ being transported.
4. Storage: The captured $CO_2$ is injected into a suitable underground formation, such as a saline aquifer or depleted oil or gas reservoir, for permanent storage.
5. Monitoring and verification: It is important to monitor and verify that the $CO_2$ remains stored in the underground formation to ensure that it is not released back into the atmosphere. This may involve using monitoring equipment and conducting periodic checks to ensure that the $CO_2$ is being stored safely and securely.

In the recent years, much research work has been done in the domain of CCUS, be it a review on capture, storage, transportation, and utilization technologies [1–7,20], policy making, deployment of CCUS pathways, risks and uncertainties associated with CCUS [2,6], determining potential storage sites [18,20], use of ML algorithms in determining the storage capacity [10,11] or analyzing the effect of different parameters on $CO_2$ sequestration [12]. However, none of the previously published work focuses on the tracking and analyzing of the patents filed in the domain of CCUS technologies, especially in BSR countries. To address this research gap in this paper, we present an analytical analysis of the patent landscape related to carbon capture, utilization, and storage (CCUS) activities in the Baltic Sea region (BSR) countries. These countries include Lithuania, Latvia, Estonia, Finland, Denmark, Sweden, Russia, Poland, and Norway [13–17]. To conduct the analysis, searches were conducted to identify patents related to CCUS in the BSR countries between 2000 and 2020. The technologies investigated mainly focus on $CO_2$ storage, monitoring, utilization, and transport. A large number of patent families were analyzed, and the results of this analysis are presented. Top of Form.

The paper is organized as follows: Introduction is presented in Section 1. Technologies used for carbon capture are presented in Section 2. $CO_2$ usage technologies are discussed at high level in Section 3. Section 4 introduces the methodology used for conducting the patent searches. Patent classification scheme is presented in Section 5 and analysis of the patents and analytical findings derived from patent analysis is presented in Section 6. Section 7 details the patent landscape trends. Investment trends based on patent analysis for some of the key industry players working in CCUS are presented in Section 8. Challenges related

to CCUS are covered in Section 9. A short discussion on future of CCUS is presented in Section 10. Finally, summary and conclusions are presented in Section 11.

### 2. CO$_2$ Capture Technologies: An Overview

Carbon dioxide (CO$_2$) capture technologies are used to separate or capture CO$_2$ from flue gas streams in industrial processes. The most widely used and advanced technologies are chemical absorption and physical separation, which use solvents or differences in physical properties to separate CO$_2$ from other gases. Other technologies include membrane separation and looping cycles such as chemical looping or calcium looping, which have the potential to be more energy-efficient and cost-effective in the future, but are still in the early stages of development. The most suitable CO$_2$ capture technology will depend on the specific application and circumstances. Factors to consider when choosing a CO$_2$ capture technology include the concentration of CO$_2$ in the gas stream, the size and type of the facility, the cost of the technology, and the availability of energy for the capture process. The various technologies are described in Table 1.

**Table 1.** CCUS capture technology.

| Capture Technology | Overview | Current Status |
|---|---|---|
| Chemical absorption | Chemical absorption involves the use of a solvent to absorb CO$_2$ from a gas stream, after which the solvent is regenerated and the CO$_2$ is separated. The most commonly used solvents are amines, which react with CO$_2$ to form a stable compound. The CO$_2$ is then released from the solvent by heating. Chemical absorption is a widely used and advanced technology, but it can be energy-intensive and costly. | Chemical absorption is a widely used and advanced technology for CO$_2$ capture. However, it is energy-intensive and costly, and there are ongoing efforts to improve its efficiency and reduce costs. |
| Physical separation | Physical separation technologies, such as cryogenics and adsorption, use the difference in physical properties between CO$_2$ and other gases to separate them. In cryogenic separation, the gas stream is cooled until the CO$_2$ condenses and can be collected. Adsorption technologies use a solid material to trap CO$_2$ from the gas stream, after which the CO$_2$ can be released by heating or pressure change. Physical separation technologies are widely used and advanced, but they can be energy-intensive and costly. | Physical separation technologies are widely used and advanced for CO$_2$ capture. However, they can be energy-intensive and costly, and there are ongoing efforts to improve their efficiency and reduce costs. |
| Oxy-fuel separation | Oxy-fuel separation involves burning a fuel with oxygen instead of air, resulting in a flue gas that is mostly CO$_2$ and water vapor. The CO$_2$ can then be separated from the water vapor by condensation. Oxy-fuel separation is a promising technology, but it is not yet widely used due to technical and economic challenges. | Oxy-fuel separation is a promising technology for CO$_2$ capture, but it is not yet widely used due to technical and economic challenges. There are ongoing efforts to improve its efficiency and reduce costs. |
| Membrane separation | Membrane separation technologies use a semi-permeable membrane to separate CO$_2$ from a gas stream based on the difference in gas permeability. The CO$_2$ is absorbed by the membrane, while other gases pass through. Membrane separation can be more energy-efficient than other technologies, but it is not yet widely used for CO$_2$ capture due to technical and economic challenges. | Membrane separation is a promising technology for CO$_2$ capture, but it is not yet widely used due to technical and economic challenges. There are ongoing efforts to improve its efficiency and reduce costs. |
| Calcium looping | Calcium looping is a variation of chemical looping that uses calcium oxide (CaO) as the solid material for absorbing CO$_2$. The CaO is heated to release the CO$_2$, which can be collected. The CaO is then regenerated by reacting with calcium carbonate (CaCO$_3$), which is abundant and inexpensive. Calcium looping has the potential to be more energy-efficient and cost-effective. | Technology has been tested on some pilot projects. |
| Chemical looping | Chemical looping involves the use of a solid material, such as iron oxide, to absorb CO$_2$ from a gas stream. The solid material is then heated to release the CO$_2$, which can be collected. The solid material is then regenerated and ready to absorb more CO$_2$. Chemical looping has the potential to be more energy-efficient than other technologies, but it is still in the early stages of development. | Technology has been tested on some limited pilot projects |
| Direct separation | Direct air separation involves capturing CO$_2$ directly from the ambient air using an adsorbent material. The CO$_2$ is then released from the adsorbent by heating or pressure change. | Energy intensive process but pilot projects have been in operation in Europe and the USA. |
| Supercritical CO$_2$ power cycles | This technology is still in the early stages of development, with only a few small-scale demonstrations in operation. However, it has the potential to significantly increase the efficiency of power generation compared to traditional thermal power plants, as well as reducing CO$_2$ emissions by capturing and storing the CO$_2$ produced. It is particularly well-suited for use with renewable energy sources, such as solar or nuclear, which produce heat that can be used to drive the supercritical CO$_2$ cycle. | Trigen project developed by Maersk Oil. |

### 3. CO$_2$ Usage Technologies: An Overview

Carbon dioxide (CO$_2$) has the potential to be used as an input for a variety of products and services. Applications for CO$_2$ use can be divided into two categories: direct use,

where the $CO_2$ is not chemically modified (non-conversion), and the transformation of $CO_2$ into a useful product through chemical and biological processes (conversion). In this paper, we compare the potential scale and cost of different $CO_2$ utilization pathways. Overall, $CO_2$ utilization has the potential to operate at a large scale and at low cost, which could make it a lucrative business opportunity in the future.

- **$CO_2$ chemicals:** $CO_2$ can be used as a feedstock to produce chemicals such as methanol, formic acid, and urea. These chemicals have a variety of industrial and consumer applications, including the production of plastics, solvents, and fertilizers.
- **$CO_2$ fuels:** $CO_2$ can be used to produce fuels such as syngas and methanol, which can be burned to generate energy. These fuels can be produced from $CO_2$ through processes such as carbon capture and utilization (CCU) or carbon capture and storage (CCS).
- **Microalgae:** Microalgae are tiny aquatic organisms that can be used to produce biofuels and other products. They can absorb $CO_2$ from the air and convert it into biomass through photosynthesis.
- **Concrete building materials:** $CO_2$ can be used to produce low-carbon concrete building materials, which can reduce the carbon footprint of the construction industry.
- **$CO_2$-enhanced oil recovery (EOR):** $CO_2$ can be injected into depleted oil reservoirs to help extract remaining oil, a process known as enhanced oil recovery (EOR). This can help extend the life of oil fields and reduce the need for new oil exploration.
- **Bioenergy with carbon capture and storage:** Bioenergy with carbon capture and storage (BECCS) involves using biomass to generate energy and capturing the $CO_2$ emissions produced in the process. The $CO_2$ can then be stored underground to mitigate climate change.
- **Soil carbon sequestration:** Soil carbon sequestration involves storing $CO_2$ in the soil through practices such as reducing tillage, adding organic matter, and planting cover crops. This can help reduce $CO_2$ in the atmosphere and improve soil health.
- **Biochar:** Biochar is produced by "pyrolyzing" biomass, which is plant material that has been burned at high temperatures in a low-oxygen environment. When applied to agricultural soils, biochar has the potential to increase crop yields by 10%. However, it can be difficult to consistently produce a high-quality biochar product or predict how it will react with different soils. We estimate that biochar could utilize 0.2 to 1 Gt of $CO_2$ in 2050, at costs of approximately \$65 per ton of $CO_2$.

## 4. Patent Search Methodology

In this paper, the technologies used in carbon removal are analyzed by tracking patents filed in the field of carbon removal technologies. A patent search was conducted to identify patents related to carbon capture, utilization, and storage (CCUS) technology. The patent search was conducted using the Orbit database [31], with the following search strings (Table 2).

The searches resulted in a total of 3299 patent families related to CCUS. A relevancy analysis was done to identify which of the patents are directly related to CCUS and it resulted in 497 patent families. Identified relevant patents have been categorized in a classification scheme described in the next section. The searches were conducted in June 2022 and results include patent/patent applications published until the time.

Table 2. Patent search.

| Sl No | Query | Concept | Result(s) |
|---|---|---|---|
| 1 | ((((Carbon or (carbon d dioxide) or CO$_2$) 3d (captur+ or storage or storing or sequestration or evaluation or trap+ or transport+ or utliz+)) or (ccus or ccs)))/ti/ab/clms/desc/odes/tx | CO$_2$ capture/storage in Full text | 154,439 |
| 2 | Prd ≥ 2000 | Year restriction 2000–2020 | 45,445,454 |
| 3 | (Lt or lv or ee or fi or dk or se or ru or pl or no or is)/pn | Country restriction | 2,611,315 |
| 4 | 1 and 2 and 3 | | 7578 |
| 5 | ((Carbon or (carbon d dioxide) or CO$_2$) 3d (captur+ or stor+ or sequestration or evaluation or trap+ or transport+ or utliz+))/ti/ab/clms/desc/odes/tx | CO$_2$ capture/storage in Full text | 108,111 |
| 6 | 5 and 2 and 3 | CO$_2$ capture/storage in FT and Priority Date and CC | 4564 |
| 7 | (Y02c+ or y02e+ or y02+ or g01n+ or g01v+)/ipc/cpc | Classification | 5,482,625 |
| 8 | 6 and 7 | | 2133 |
| 9 | 4 and 7 | | 2783 |
| 10 | ((((Carbon or (carbon d dioxide) or CO$_2$) 3d (captu+ or grab+ or aquir+ or trap+)))/ti/ab/clms | CO$_2$ capture in Title abstract & Claims | 7943 |
| 11 | 10 and 2 and 3 | CO$_2$ capture in Title, abstract & Claims and Priority Date and CC | 399 |
| 12 | ((Carbon or (carbon d dioxide) or CO$_2$) 3d (stor+ or conserv+ or reserv+ or sequestration))/ti/ab/clms | CO$_2$ capture in Title abstract & Claims | 27,727 |
| 13 | 12 and 2 and 3 | CO$_2$ storage in Title, abstract & Claims and Priority Date and CC | 617 |
| 14 | ((((Carbon or (carbon d dioxide) or CO$_2$) 3d (transport+ or ship+ or pipe+)))/ti/ab/clms | CO$_2$ Transport in Title abstract & Claims | 44,291 |
| 15 | 14 and 2 and 3 | CO$_2$ Transport in Title abstract & Claims | 843 |
| 16 | ((((Carbon or (carbon d dioxide) or CO$_2$) 3d (captur+ or stor+ or sequestration or evaluation or trap+ or transport+ or utliz+)) 5d (monitor+ or surve+ or controll+))/ti/ab/clms/tx | CO$_2$ capture/storage in Full Text with monitoring | 1898 |
| 17 | 16 and 2 and 3 | CO$_2$ capture/storage in Full Text with monitoring and Priority Date and CC | 62 |
| 18 | 11 or 13 or 15 or 17 | | 1749 |
| 19 | 18 or 8 | Without CPC | 3299 |
| 20 | 18 and 7 | | 728 |
| 21 | 20 or 8 | With CPC | 2278 |

## 5. Patent Classification Scheme

Analysis of 497 relevant patent families is not an easy task. In order to aid the analysis, it was decided that the CCUS patent landscape should be categorized based on the following schemas. The schema (1), primary classification, focuses on CCUS type, i.e.,

capture technologies, storage, utilization, and transport (Figure 2). Schema (2), secondary classification, focuses on applications of CCUS (Figure 3). Based on the searches, four broad categories of patents were decided for further analysis. The four categories, based on primary classification scheme, are further described below:

- **Capture**—The CO$_2$ is separated from other gases produced in industrial processes, such as those at coal and natural-gas-fired power generation plants, or steel or cement factories.
- **Transport**—The CO$_2$ is then compressed and transported via pipelines, road transport or ships to a site for storage.
- **Storage**—Finally, the CO$_2$ is injected into rock formations deep underground for permanent/temporary storage.
- **Utilization**—The carbon captured could be re-used in industrial processes by converting it into, for example, plastics, concrete, or biofuel.

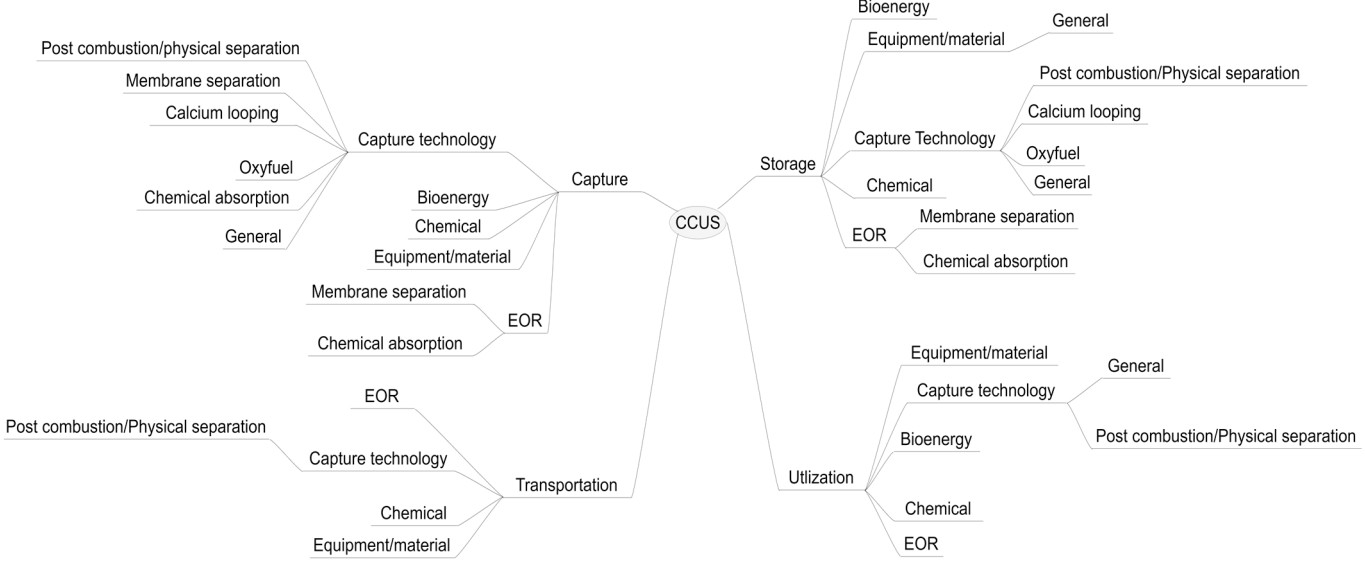

**Figure 2.** Schema (1) CCUS primary classification.

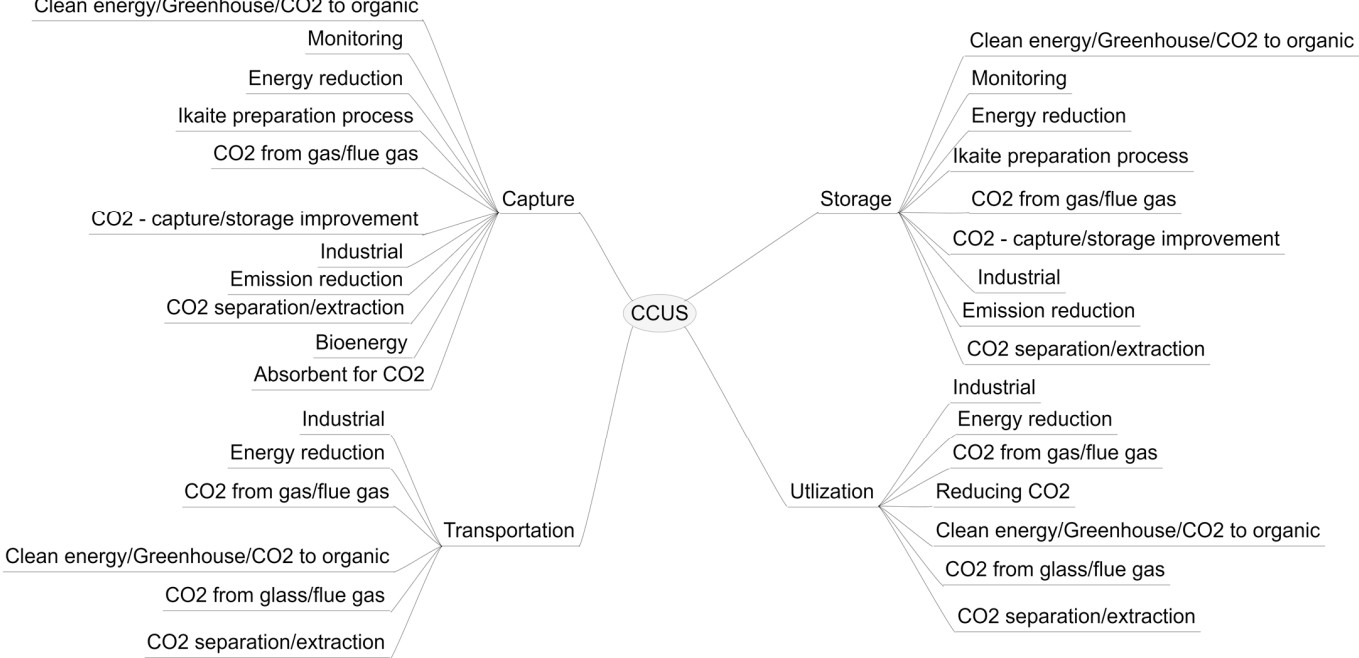

**Figure 3.** Schema (2) CCUS secondary classification.

## 6. Patent Landscape Trends—Non-Technology Trends

A total of 497 patent families are found relevant in the analysis related to CCUS technology. Below is a chart representing the earliest priority filing (first patent application date) trend. It is important to highlight that all the patents filed in 2021 might not have been published until the date of study, which may have some impact on the trends reported in this section.

From Figure 4, it is observed that 2009 has the greatest number of IP activity for CCUS for both applications and grants. Exponential growth in patent filing since 2005–2009 has an increasing trend for CCUS activities, while 2010–2015 has an exponential decreasing trend for CCUS activities.

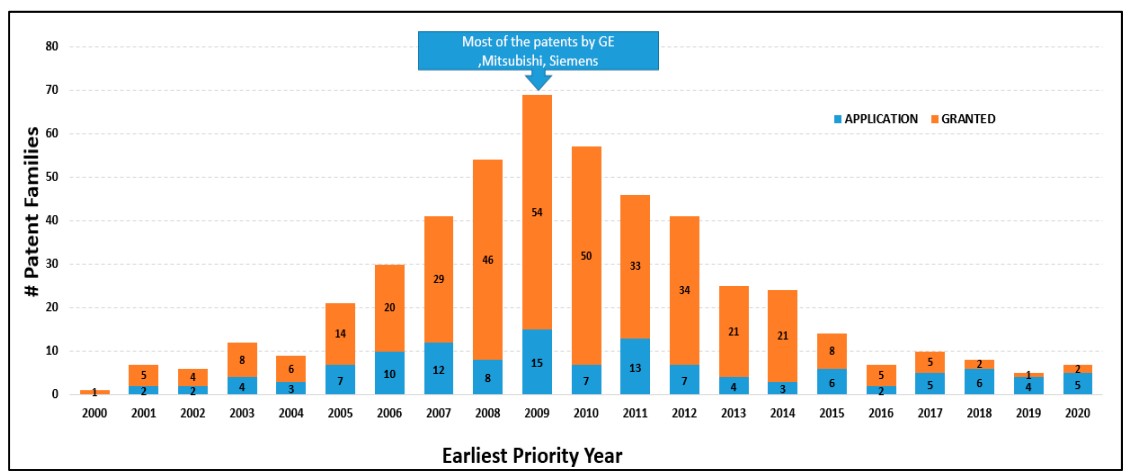

**Figure 4.** Patent filling—first application year.

CCUS technology has found attention all over northern and eastern European countries. Figure 5 shows the geographical representation of patent filing across LT (Lithuania), LV (Latvia), EE (Estonia), FI (Finland), DK (Denmark), SE (Sweden), RU (Russia), PL (Poland), NO (Norway), IS (Iceland), ES (Spain) and the top assignees.

| Assignees | Russia | Poland | Norway | Spain | Denmark | Germany | Austria | Hungary | Italy |
|---|---|---|---|---|---|---|---|---|---|
| GE | 42 | 12 | 3 | 5 | 5 | 2 | | | |
| MITSUBISHI | 23 | | 10 | | 10 | 3 | | | |
| SIEMENS | 14 | 9 | 1 | 4 | 1 | 4 | | | |
| IFP | 6 | 7 | 5 | 4 | 3 | 4 | 5 | 1 | 1 |
| KANSAI ELECTRIC POWER | 10 | | 5 | | 4 | 1 | | | |
| AIR PRODUCTS & CHEMICALS | 2 | 12 | 1 | 5 | 1 | 2 | 2 | | |
| AKER CARBON CAPTURE | 2 | 8 | 10 | 3 | 1 | | | | |
| IHI | | 11 | 1 | 9 | | | | | |
| AIR LIQUIDE | 3 | 9 | | 4 | | 2 | 3 | | |

**Figure 5.** Patent publication countries with assignees (Industry). Color intensity is proportionate to higher figure i.e., the more green it is the higher the number of assignees in each country it represents.

Russia and Poland are leading the research and patent filing in the CCUS domain among the Baltic Sea region countries. Among the companies, GE has more filings in RU and PL, and IFP has consistent filing in the north-Eastern European countries.

### 6.1. Patent Filing Trend across Baltic Sea Region Countries

The patent filing trend (Figure 6) below represents country-wise trends across priority years. Figure 6 represents all the north-eastern European countries and Figure 7 represents

worldwide filing across filing years. The highlighted sections represent the north-eastern European countries (Figure 7).

1st Application year

| Country | 2002 | 2003 | 2004 | 2005 | 2006 | 2007 | 2008 | 2009 | 2010 | 2011 | 2012 | 2013 | 2014 | 2015 | 2016 | 2017 | 2018 | 2019 | 2020 | 2021 | 2022 |
|---|---|---|---|---|---|---|---|---|---|---|---|---|---|---|---|---|---|---|---|---|---|
| Russia |  | 1 | 5 | 7 | 8 | 18 | 21 | 26 | 46 | 25 | 23 | 19 | 13 | 12 | 6 | 4 | 5 | 2 | 2 |  |  |
| Poland | 1 | 1 | 5 | 2 | 6 | 8 | 27 | 17 | 17 | 25 | 19 | 14 | 12 | 10 | 4 | 3 | 4 |  |  |  |  |
| Norway | 2 | 3 | 5 | 7 | 8 | 11 | 8 | 18 | 11 | 7 | 10 | 5 | 5 | 3 | 1 | 1 | 2 | 3 | 3 |  |  |
| Spain | 2 | 1 | 5 | 1 | 5 | 8 | 16 | 13 | 12 | 17 | 11 | 8 | 7 | 2 | 3 | 1 | 2 |  |  |  |  |
| Denmark | 3 |  | 4 | 4 | 2 | 4 | 7 | 12 | 10 | 10 | 8 | 8 | 6 |  | 3 | 1 | 1 | 1 | 3 |  |  |
| Germany | 3 | 3 | 6 | 2 | 6 | 5 | 7 | 8 | 3 | 4 | 1 | 2 | 1 |  | 1 |  |  |  |  |  |  |
| Austria | 1 | 3 | 6 | 2 | 5 | 2 | 6 | 7 | 1 | 3 |  |  |  |  |  |  |  |  |  |  |  |
| Hungary |  |  |  |  |  | 1 | 1 | 2 |  | 2 | 3 | 1 | 1 |  | 2 |  |  |  |  |  |  |
| Finland |  | 1 |  |  |  | 1 | 1 | 2 | 2 |  | 2 |  |  |  |  |  |  |  |  |  |  |
| Lithuania |  |  |  |  |  | 1 | 1 |  |  | 2 | 1 |  |  |  |  |  | 1 |  |  |  |  |
| Iceland |  |  |  |  | 1 |  |  |  |  |  |  |  |  |  |  |  |  |  |  |  |  |

**Figure 6.** Patent filing trend across Northern and Eastern European countries. The figure uses two colors to represent high and low numbers i.e., the more green represents high number and more red indicates a lower figure of patent filling trend across the countries.

1st Application year

| Publication country | 2002 | 2003 | 2004 | 2005 | 2006 | 2007 | 2008 | 2009 | 2010 | 2011 | 2012 | 2013 | 2014 | 2015 | 2016 | 2017 | 2018 | 2019 | 2020 | 2021 | 2022 |
|---|---|---|---|---|---|---|---|---|---|---|---|---|---|---|---|---|---|---|---|---|---|
| Europe | 6 | 5 | 12 | 9 | 22 | 28 | 51 | 53 | 55 | 52 | 47 | 38 | 24 | 10 | 11 | 5 | 7 | 7 |  |  |  |
| United States of America | 6 | 8 | 11 | 13 | 26 | 29 | 45 | 53 | 70 | 52 | 47 | 47 | 31 | 25 | 17 | 12 | 8 | 8 | 6 | 3 | 1 |
| Canada | 5 | 2 | 8 | 9 | 10 | 21 | 34 | 37 | 47 | 42 | 24 | 25 | 16 | 14 | 7 | 3 | 5 | 3 | 1 |  |  |
| China | 2 | 1 | 5 | 3 | 14 | 18 | 35 | 29 | 44 | 39 | 32 | 29 | 16 | 14 | 5 | 3 | 5 | 6 | 1 |  |  |
| Australia | 6 | 5 | 5 | 6 | 7 | 15 | 39 | 34 | 37 | 37 | 24 | 14 | 14 | 14 | 10 | 3 | 4 | 2 | 2 |  |  |
| Japan | 4 | 4 | 8 | 8 | 13 | 14 | 37 | 34 | 29 | 29 | 28 | 18 | 14 | 8 | 6 | 4 | 4 | 3 |  |  |  |
| Russia |  | 1 | 5 | 7 | 8 | 18 | 21 | 26 | 46 | 25 | 23 | 19 | 13 | 12 | 6 | 4 | 5 | 2 | 2 |  |  |
| India |  |  | 3 | 2 | 3 | 4 | 13 | 18 | 30 | 32 | 29 | 25 | 18 | 14 | 13 | 5 | 4 | 3 | 4 |  |  |
| Poland | 1 | 1 | 5 | 2 | 6 | 8 | 27 | 17 | 17 | 25 | 19 | 14 | 12 | 10 | 4 | 3 | 4 |  |  |  |  |
| Brazil | 1 |  | 4 | 2 | 7 | 9 | 19 | 16 | 39 | 22 | 15 | 14 | 8 | 6 | 3 | 1 | 4 | 2 | 1 |  |  |
| Korea |  |  | 3 | 2 | 3 | 7 | 26 | 18 | 21 | 24 | 21 | 11 | 8 | 7 | 1 |  |  | 3 | 2 |  |  |
| Norway | 2 | 3 | 5 | 7 | 8 | 11 | 8 | 18 | 11 | 7 | 10 | 5 | 5 | 3 | 1 | 1 | 2 | 3 | 3 |  |  |
| Spain | 2 | 1 | 5 | 1 | 5 | 8 | 16 | 13 | 12 | 17 | 11 | 8 | 7 | 2 | 3 | 1 | 2 |  |  |  |  |
| South Africa |  |  | 3 | 2 | 0 | 2 | 4 | 4 | 13 | 22 | 12 | 13 | 7 | 4 | 2 | 1 |  |  |  |  |  |
| Denmark | 3 |  | 4 | 4 | 2 | 4 | 7 | 12 | 10 | 10 | 8 | 8 | 6 |  | 3 | 1 | 1 | 1 | 3 |  |  |
| Mexico | 1 |  | 3 | 2 | 2 | 4 | 11 | 10 | 14 | 12 | 9 | 3 | 2 | 2 | 1 | 1 |  |  |  |  |  |
| Germany | 3 | 3 | 6 | 2 | 6 | 5 | 7 | 8 | 3 | 4 | 1 | 2 | 1 |  | 1 |  |  |  |  |  |  |
| Austria | 1 | 3 | 6 | 2 | 5 | 2 | 6 | 7 | 1 | 3 |  |  |  |  |  |  |  |  |  |  |  |
| Taiwan |  |  | 1 | 2 | 2 | 1 | 2 | 3 | 6 | 9 | 4 | 2 | 1 | 6 |  |  |  |  |  |  |  |

**Figure 7.** Patent filing trend worldwide. The figure uses two colors to represent high and low numbers i.e., the more green represents high number and more red indicates a lower figure of patent filling trend across the countries.

A sudden surge in patent filing in Russia can be seen since 2007 onwards. Continued interest in patent filing is potentially due to increased application-based industries coming up in the region. Continuous patent filing in Russia could also be due to an increased interest of various big player companies like GE, Mitsubishi, Siemens, etc., as well as start-ups in the technology. PL, NO, ES, and DK have more filings from 2006 to 2014. EP and US have the highest filings followed by CA and CN, demonstrating that while the US, EP, CA, and CN are the countries that provide knowledge through patents, other countries, such as Russia and Poland, are the main countries that receive technology. Another case is China, where the amount of material published is similar to the amount generated since their inventions are not transferred worldwide but remain in the country.

*6.2. Top Assignees*

Figure 8 below shows the chart of top assignees identified in the search. GE has the highest number of publications followed by Mitsubishi and Siemens. GE is helping Poland advance its energy strategy to cut carbon and promote renewables. GE and E energy partnered to deliver a 68.9 MW wind farm in Lithuania with Cypress turbines.

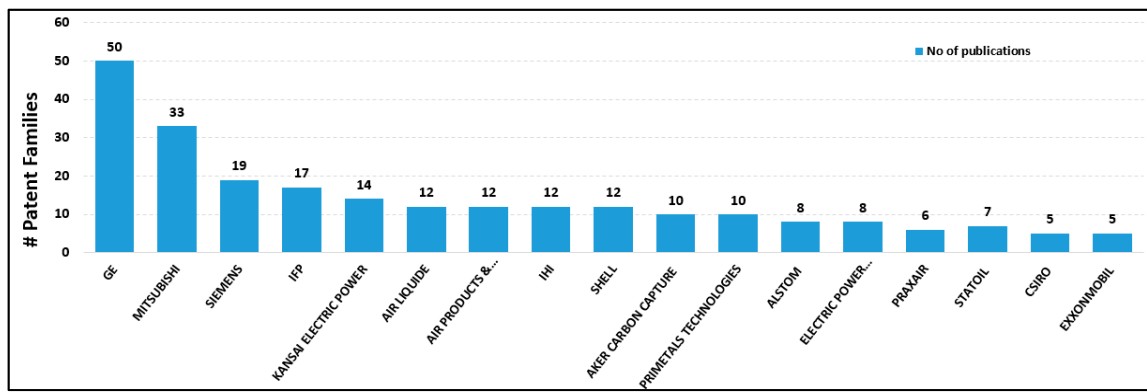

**Figure 8.** Top assignees (Industry).

It is also important to analyze the current legal status of the patents filed by some of the technology companies (Figure 9); the figure represents the legal status of all the patents. It can be seen that 85% of 497 relevant patent families are alive and active, with GE having around 78% of its families alive.

**Country wise- Legal Status**

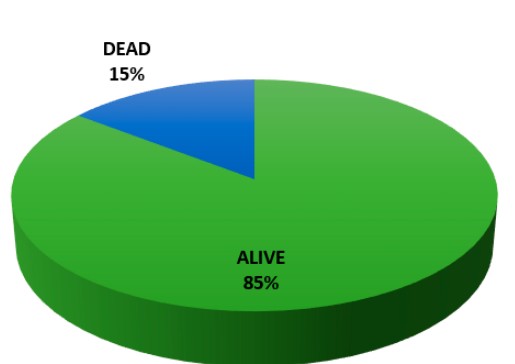

| Assignees | Alive | Dead |
|---|---|---|
| GE | 41 | 9 |
| MITSUBISHI | 29 | 4 |
| KANSAI ELECTRIC POWER | 11 | 3 |
| SIEMENS | 16 | 3 |
| AIR LIQUIDE | 11 | 1 |
| AIR PRODUCTS & CHEMICALS | 11 | 1 |
| IHI | 12 | |
| IFP | 9 | 8 |
| AKER CARBON CAPTURE | 10 | |
| PRIMETALS TECHNOLOGIES | 10 | |
| ALSTOM TECHNOLOGY | 6 | 2 |
| ELECTRIC POWER DEVELOPMENT | 8 | |

**Figure 9.** Legal status of the patent families.

## 7. Patent Landscape Trends—Technology Trends

Next, we analyze the patents based on technology trends. Figure 10 below represents classification of the technology trends of relevant patents. Top 10 classes have been represented in Figure 10 below, where description of the classes is given in Table 3. This graph shows the distribution of the main classification codes contained in the patent portfolio and can help to identify the subject areas in which the applicant seeks protection. Y02C is the top classification code, which is related to CAPTURE, STORAGE, SEQUESTRATION OR DISPOSAL OF GREENHOUSE GASES and followed by B01D, which is related to Separation.

Figure 11 shows the distribution of the four major categories of CCUS technologies and its distribution among the Baltic Sea region countries. It can be seen that RU (Russia) has the greatest number of patents/publications in $CO_2$ capture, storage, utilization, and transport followed by PL (Poland) and NO (Norway). $CO_2$ capture is the most explored technology/CCUS type. Capture technology along with storage is the most common.

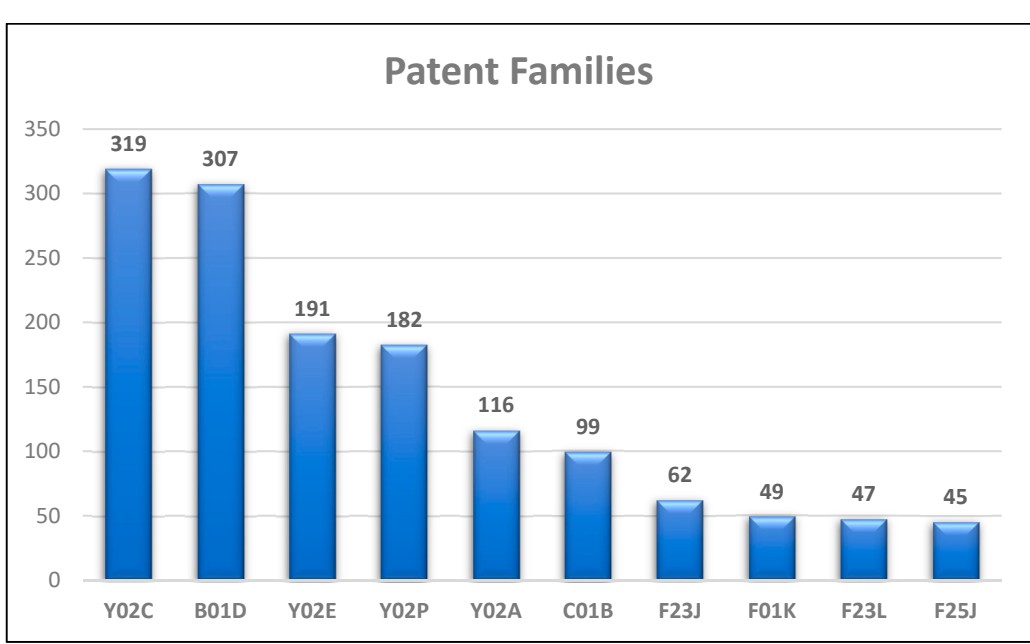

**Figure 10.** Main classification categories (CPC).

**Table 3.** Description of main CPC.

| | |
|---|---|
| Y02C | capture, storage, sequestration or disposal of greenhouse gases |
| B01D | separation |
| Y02E | reduction of greenhouse gas [ghg] emissions, related to energy generation, transmission or distribution |
| Y02P | climate change mitigation technologies in the production or processing of goods |
| Y02A | technologies for adaptation to climate change |
| C01B | non-metallic elements; compounds thereof |
| F23J | removal or treatment of combustion products or combustion residues; flues |
| F01K | steam engine plants; steam accumulators; engine plants not otherwise provided for; engines using special working fluids or cycles |
| F23L | supplying air or non-combustible liquids or gases to combustion apparatus in general |
| F25J | liquefaction, solidification or separation of gases or gaseous or liquefied gaseous mixtures by pressure and cold treatment |

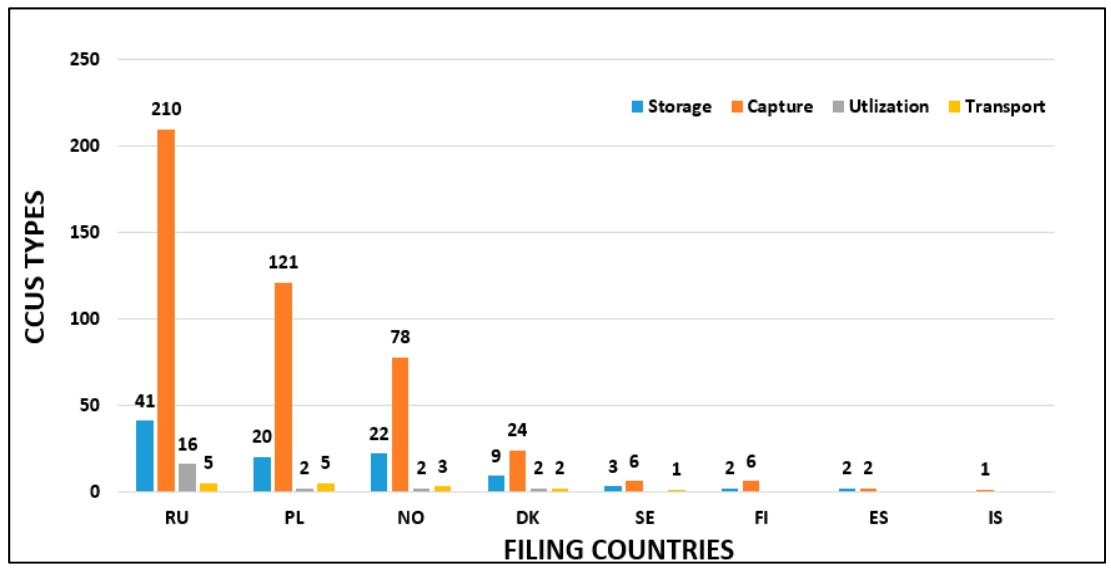

**Figure 11.** CCUS types—countries (Level 1 category).

Now, if we analyze the priority year of the technology patents, we can see from Figures 12 and 13 that 2009 has the most number of filing for capture. Capture technology has been prominent from 2003 to 2013. It can also be seen that year 2009 has the highest number of filings in all CCUS technology types. It can also be observed that 2005–2014 was advantageous for carbon capture, followed by storage. Additionally, it can be observed that there is a decreasing trend in patent filings since 2016, which has continued through 2020.

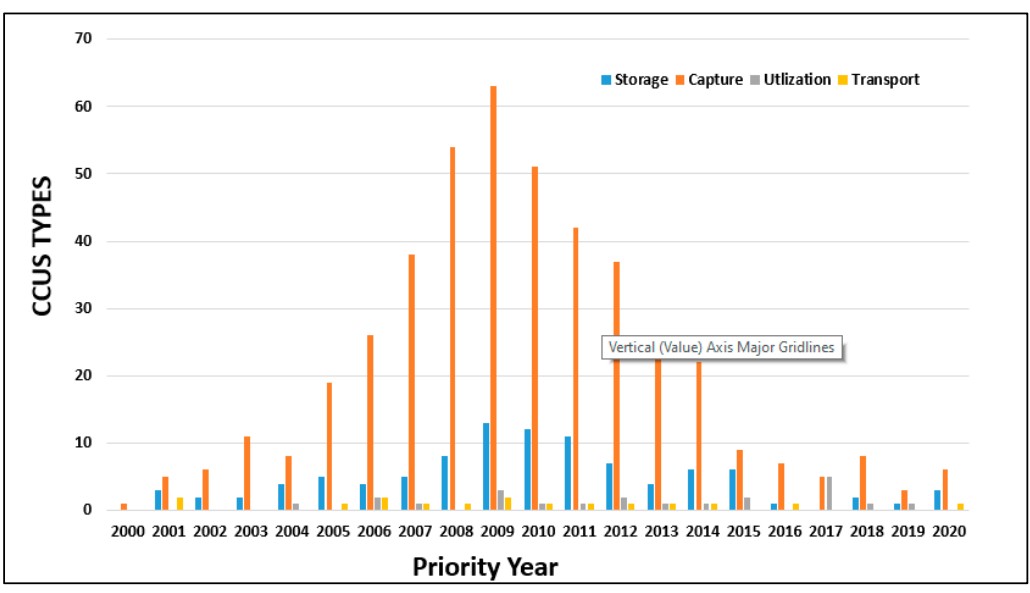

**Figure 12.** CCUS types—priority year (bar chart).

| EPR | Storage | Capture | Utlization | Transport |
|---|---|---|---|---|
| 2000 | | 1 | | |
| 2001 | 3 | 5 | | 2 |
| 2002 | 2 | 6 | | |
| 2003 | 2 | 11 | | |
| 2004 | 4 | 8 | 1 | |
| 2005 | 5 | 19 | | 1 |
| 2006 | 4 | 26 | 2 | 2 |
| 2007 | 5 | 38 | 1 | 1 |
| 2008 | 8 | 54 | | 1 |
| 2009 | 13 | 63 | 3 | 2 |
| 2010 | 12 | 51 | 1 | 1 |
| 2011 | 11 | 42 | 1 | 1 |
| 2012 | 7 | 37 | 2 | 1 |
| 2013 | 4 | 23 | 1 | 1 |
| 2014 | 6 | 22 | 1 | 1 |
| 2015 | 6 | 9 | 2 | |
| 2016 | 1 | 7 | | 1 |
| 2017 | | 5 | 5 | |
| 2018 | 2 | 8 | 1 | |
| 2019 | 1 | 3 | 1 | |
| 2020 | 3 | 6 | | 1 |

**Figure 13.** CCUS types—priority year analysis. The figure uses two colors to represent high and low numbers i.e., the more green represents a higher number and more red indicates a lower figure. Blue box represents the highest numbers of CCUS types (capture).

Detailed analysis of the technology patents shown in Figure 14, shows that capture technology has the greatest number of patents/publications, general and post combustion being the top categories followed by Oxyfuel. Again, it can be seen that RU (Russia) seems to stand out as the top country in storage, capture, utilization, and transport.

| CCUS TYPE | Storage | | | | | | | Capture | | | | | | | | Utlization | | | | Transport | | | | |
|---|---|---|---|---|---|---|---|---|---|---|---|---|---|---|---|---|---|---|---|---|---|---|---|---|
| Countries | DK | ES | FI | NO | PL | RU | SE | DK | ES | FI | IS | NO | PL | RU | SE | DK | NO | PL | RU | DK | NO | PL | RU | SE |
| **Bioenergy** | | 1 | | | 1 | 1 | | | | 1 | | | | 1 | | 1 | | | | | | | | |
| Others | | 1 | | | 1 | 1 | | | | 1 | | | | 1 | | 1 | | | | | | | | |
| **Capture Technology** | 5 | | 2 | 6 | 11 | 20 | 2 | 12 | 1 | 3 | 1 | 48 | 80 | 126 | 3 | | | 2 | 7 | 1 | 2 | 3 | 2 | |
| Absorption | | | | | | | | 1 | | | | 8 | 2 | 15 | | | | | | | | | | |
| Calcium looping | 1 | | | | 1 | | 1 | 1 | 1 | | | 1 | 3 | 1 | 1 | | | | | | | | | |
| Chemical absorption | | | | | | | | 2 | | | | 1 | 4 | 7 | | | | | 2 | | | | | |
| General | 4 | | 2 | 4 | 5 | 10 | 1 | 6 | | 2 | 1 | 22 | 33 | 57 | 1 | | | | 4 | 1 | 1 | 2 | | |
| Membrane Separation | | | | | | | | | | | | 3 | 2 | 7 | 1 | | | | | | | | | |
| Others | | | | 1 | | | | | | | | 3 | 1 | | | | 1 | | | | | | | |
| Oxyfuel | | | | 1 | 2 | 6 | | | | | 1 | 3 | 19 | 13 | | | | | 1 | | 1 | | 1 | |
| Post Combustion/Physical separation | | | | 1 | 2 | 2 | | 2 | | | | 9 | 12 | 23 | | | 1 | | | | | 1 | 1 | |
| Pre combustion | | | | | | 2 | | | | | | 1 | 2 | 2 | | | | | | | | | | |
| **Chemical** | | | | 1 | 4 | 4 | | 2 | | | | 4 | 11 | 13 | 1 | | | | 5 | | | 1 | 1 | |
| Absorption | | | | | | | | | | | | | | 2 | | | | | | | | | | |
| Chemical absorption | | | | | | | | 1 | | | | 3 | 8 | 8 | | | | | 1 | | | | | |
| General | | | | 1 | 4 | 4 | | | | | | | 2 | 1 | 1 | | | | | | | 1 | | |
| Membrane Separation | | | | | | | | | | | | 1 | | | | | | | | | | | | |
| Others | | | | | | | | 1 | | | | | | 1 | | | | | 4 | | | | 1 | |
| Post Combustion/Physical separation | | | | | | | | | | | | | 1 | 1 | | | | | | | | | | |
| **EOR** | | | | 12 | 2 | 4 | | 2 | | | | 12 | 3 | 7 | | 1 | 2 | | 1 | 1 | | | | |
| Chemical absorption | | | | 1 | | | | | | | | 1 | | | | | | | | | | | | |
| General | | | | 9 | 1 | 3 | | 1 | | | | 6 | 2 | 6 | | 1 | | | | | | | | |
| Membrane Separation | | | | 2 | | | | | | | | 2 | | | | | | | | | | | | |
| Others | | | | 1 | | | | 1 | | | | 2 | | | | | 2 | | 1 | 1 | | | | |
| Oxyfuel | | | | | | 1 | | | | | | 1 | | 1 | | | | | | | | | | |
| Post Combustion/Physical separation | | | | | | | | | | | | | 1 | | | | | | | | | | | |
| **Equipment/material** | 4 | 1 | | 4 | 2 | 11 | 1 | 8 | | 3 | | 15 | 27 | 60 | 2 | | | | 3 | | 1 | 1 | 2 | 1 |
| Absorption | | | | | | | | 1 | | | | 2 | 1 | 6 | | | | | | | | | | |
| Chemical absorption | | | | | | | | | | | | 1 | | 1 | | | | | | | | | | |
| General | 3 | | | 1 | 1 | 5 | 1 | 2 | | 1 | | 3 | 5 | 21 | | | | | 1 | | 1 | | 1 | |
| Membrane Separation | | | | | | 2 | | | | | | 1 | 1 | 3 | | | | | | | | | | |
| Others | 1 | 1 | | | | | | 4 | | 1 | | 4 | 10 | 20 | 2 | | | | 1 | | | | 1 | 1 |
| Oxyfuel | | | | | 2 | | | | | | | 1 | 2 | 5 | | | | | 1 | | | | | |
| Post Combustion/Physical separation | | | | 3 | 1 | 3 | | 1 | | 1 | | 3 | 5 | 4 | | | | | | | | 1 | | |
| Pre combustion | | | | | | | | | | | | 3 | | | | | | | | | | | | |
| **Total** | 9 | 2 | 2 | 23 | 20 | 41 | 3 | 24 | 2 | 6 | 1 | 79 | 121 | 207 | 6 | 2 | 2 | 2 | 16 | 2 | 3 | 5 | 5 | 1 |
| **Total CCUS** | 100 | | | | | | | 446 | | | | | | | | 22 | | | | 16 | | | | |

**Figure 14.** Technology focus (Primary)—countries. The figure uses two colors to represent high and low numbers i.e., the more green represents a higher number and more red indicates a lower figure in each of CCUS types and technology.

If we now focus on secondary classification and analyze the patents, from the analysis shown in Figure 15, it can be seen that $CO_2$ from gas/flue gas is the top category in capture. RU again has more patents/publications in $CO_2$ from gas/flue gas, followed by PL and NO. Capture technology has more filings when compared to the other categories. Utilization and transport are the least filed areas. It is observed that $CO_2$ from gas/flue gas has presence in capture, storage, utilization, and transport and has the greatest filings from Russia. Capture technology is the top category in capture and storage. Bioenergy and chemicals are the least explored. $CO_2$ separation/extraction and $CO_2$ from gas/flue gas are most prominent in all the divisions.

| CCUS TYPE | Storage | | | | | | | Capture | | | | | | | | Utilization | | | | Transport | | | | |
|---|---|---|---|---|---|---|---|---|---|---|---|---|---|---|---|---|---|---|---|---|---|---|---|---|
| Countries | DK | ES | FI | NO | PL | RU | SE | DK | ES | FI | IS | NO | PL | RU | SE | DK | NO | PL | RU | DK | NO | PL | RU | SE |
| **Bioenergy** | | 1 | | | 1 | 1 | | | 1 | | | | | 1 | | 1 | | | | | | | | |
| CO2 capture/storage improvement | | | | 1 | | | | | | | | | | | | | | | | | | | | |
| Energy reduction | | 1 | | | 1 | | | | 1 | | | | | 1 | | 1 | | | | | | | | |
| **Capture Technology** | 5 | | 2 | 6 | 11 | 20 | 2 | 12 | 1 | 3 | 1 | 48 | 80 | 126 | 3 | | | 2 | 7 | 1 | 2 | 3 | 2 | |
| Clean energy/Greenhouse/CO2 to organic | | | | | 2 | | | | | | | | 1 | 7 | | | | | 2 | | | | | |
| CO2 capture/storage improvement | | | | 1 | 2 | | | | | | | 2 | 2 | 4 | | | | | | | | | | |
| CO2 from gas/flue gas | 1 | | 2 | 2 | 3 | 9 | 1 | 7 | | 3 | | 27 | 38 | 68 | 1 | | | | 5 | | 1 | 1 | 2 | |
| CO2 separation/extraction | 2 | | | 2 | 2 | 5 | | 2 | 1 | | | 10 | 20 | 37 | 1 | | | 2 | | 1 | 1 | | | |
| Emission reduction | | | | 2 | 2 | | | 1 | | | 1 | 3 | 6 | 6 | | | | | | | | 1 | | |
| Energy reduction | | | | 1 | 1 | | | | | | | 2 | 4 | 1 | | | | | | | | 1 | | |
| Ikaite preparation process | 1 | | | | | 1 | | 1 | | | | | | | 1 | | | | | | | | | |
| Industrial | 1 | | | 1 | 2 | | | 1 | | | | 4 | 9 | 3 | | | | | | | | 1 | | |
| **Chemical** | | | | 1 | 4 | 4 | | 2 | | | | 4 | 11 | 13 | 1 | | | | 5 | | | 1 | 1 | |
| Absorbent for CO2 | | | | | | | | | | | | 1 | 3 | 3 | | | | | | | | | | |
| Bioenergy | | | | | | | | 2 | | | | | | 1 | | | | | | | | | | |
| Clean energy/Greenhouse/CO2 to organic | | | | 1 | 1 | | | | | | | 1 | 1 | 1 | | | | | | | | | | |
| CO2 capture/storage improvement | | | | 1 | 1 | 2 | | | | | | | 1 | | | | | | | | | | | |
| CO2 from gas/flue gas | | | | | | | | | | | | 3 | | 6 | | | | | 1 | | | | | |
| CO2 separation/extraction | | | | 2 | | | | | | | | 4 | | 1 | | | | | | | | 1 | 1 | |
| Emission reduction | | | | | 1 | | | | | | | 1 | 1 | | | | | | | | | | | |
| Industrial | | | | | | | | | | | | 1 | | | | | | | | | | | | |
| Reducing CO2 | | | | | | | | | | | | | | | | | | | 4 | | | | | |
| **EOR** | | | | 12 | 2 | 4 | | 2 | | | | 12 | 3 | 7 | | 1 | 2 | | 1 | 1 | | | | |
| Clean energy/Greenhouse/CO2 to organic | | | | 1 | | 1 | | 1 | | | | | | | | | | | | 1 | | | | |
| CO2 capture/storage improvement | | | | | 2 | 1 | | | | | | | | 1 | | | | | | | | | | |
| CO2 from gas/flue gas | | | | 2 | | 1 | | | | | | 4 | | 4 | | | | | | | | | | |
| CO2 separation/extraction | | | | 3 | | 1 | | 1 | | | | 3 | 2 | 2 | | 1 | 2 | | 1 | | | | | |
| Emission reduction | | | | 4 | | | | | | | | 3 | 1 | | | | | | | | | | | |
| Industrial | | | | 1 | | | | | | | | 1 | | | | | | | | | | | | |
| Monitoring | | | | 1 | | | | | | | | 1 | | | | | | | | | | | | |
| **Equipment/material** | 4 | 1 | | 4 | 2 | 11 | 1 | 8 | | 3 | | 15 | 27 | 60 | 2 | | | | 3 | | 1 | 1 | 2 | 1 |
| Clean energy/Greenhouse/CO2 to organic | | | | 1 | | 1 | | 1 | | 1 | | 1 | | 1 | | | | | | | | | 1 | |
| CO2 capture/storage improvement | | | | 1 | 3 | 1 | | | | | | | 2 | 1 | | | | | | | | | | |
| CO2 from gas/flue gas | 2 | | | | | 2 | | 4 | | 1 | | 6 | 10 | 29 | 1 | | | | | | | 1 | | |
| CO2 separation/extraction | | | | 1 | 1 | 2 | | 1 | | 1 | | 5 | 12 | 20 | 1 | | | | | | | | | |
| Reducing CO2 | | | | | | | | | | | | | | | | | | | 1 | | | | | |
| Emission reduction | | | | 2 | | 2 | | 1 | | | | 2 | 2 | 4 | | | | | 1 | | 1 | | | |
| Energy reduction | | | | | | | | | | | | | | 1 | | | | | | | | | | |
| Industrial | 1 | | | | | | | 1 | | | | 1 | 1 | 4 | | | | | 1 | | | | 1 | 1 |
| Monitoring | 1 | 1 | | | 2 | | | | | | | | | | | | | | | | | | | |
| **Total** | 9 | 2 | 2 | 23 | 20 | 41 | 3 | 24 | 2 | 6 | 1 | 79 | 121 | 207 | 6 | 2 | 2 | 2 | 16 | 2 | 3 | 5 | 5 | 1 |
| **Total CCUS** | 100 | | | | | | | 446 | | | | | | | | 22 | | | | 16 | | | | |

**Figure 15.** Technology focus (secondary)—countries. The figure uses two colors to represent high and low numbers i.e., the more green represents a higher number and more red indicates a lower figure in each of CCUS types and technology.

## 8. Investment Trend for Key Players

Next, we look at the investment trends of the major companies working in the field of CCUS. This analysis highlights the patent strategies and identifies new entrants or applicants who are no longer involved in this subject area. This information also helps explain the peaks in filing when a player files a significant number of applications over a short period of time (which could have an effect on the global evolution of filings). The results of this analysis are shown in Figure 16. GE has more filings in 2009 followed by Mitsubishi.

| Assignees | 2002 | 2003 | 2004 | 2005 | 2006 | 2007 | 2008 | 2009 | 2010 | 2011 | 2012 | 2013 | 2014 | 2015 | 2016 | 2017 | 2018 |
|---|---|---|---|---|---|---|---|---|---|---|---|---|---|---|---|---|---|
| GE | | | | 2 | 3 | 2 | 6 | 11 | 7 | 8 | 5 | | 5 | | 1 | | |
| MITSUBISHI | 1 | | 1 | 2 | 2 | 2 | 3 | 10 | 2 | 1 | 3 | 1 | 1 | 1 | 1 | | |
| SIEMENS | | | | | 1 | 2 | 2 | 6 | 4 | 1 | 2 | 1 | | | | | |
| IFP | | 3 | 1 | 2 | 3 | 2 | 1 | | | 1 | | | 1 | 1 | | | |
| KANSAI ELECTRIC POWER | | | 1 | 2 | 2 | 1 | 1 | 5 | 1 | | | | | 1 | | | |
| AIR LIQUIDE | | 1 | | | 1 | | 1 | 1 | 1 | 1 | 1 | 1 | 2 | | 1 | | 1 |
| IHI | | | | 1 | | | 8 | | 2 | 1 | | | | | | | |
| SHELL | 1 | 2 | 1 | | 4 | | | 2 | 1 | | | | | 1 | | | |
| AIR PRODUCTS & CHEMICALS | | | | | 1 | 1 | | 2 | 5 | 1 | 1 | | | | | | |
| AKER CARBON CAPTURE NORWAY | | | | 2 | 2 | | | 2 | | 1 | 3 | | | | | | |
| PRIMETALS TECHNOLOGIES | | | | | | | | 1 | 3 | | 3 | 1 | 1 | 1 | | | |
| ALSTOM TECHNOLOGY | | | | 1 | 1 | | | 3 | 3 | | | | | | | | |
| ELECTRIC POWER DEVELOPMENT | | | | 1 | | | | 7 | | | | | | | | | |
| STATOIL PETROLEUM | | | | | | | 1 | | 3 | 2 | | | | | | | |
| PRAXAIR TECHNOLOGY | | | | 2 | | | | 2 | | 1 | | | 1 | | | | |
| CSIRO | | | | | | | 2 | 1 | | 1 | 1 | | | | | | |
| EXXONMOBIL | | | | | | | | 4 | | | | | 1 | | | | |
| HALDOR TOPSOE | | | | | | | 1 | 2 | 1 | | | | | | | 1 | |
| 8 RIVERS CAPITAL | | | | | | | | | | 2 | 2 | | | | | | |
| BASF | | | | | | | | 2 | | | 1 | | 1 | | | | |
| BP | | | | | | 1 | 2 | 1 | | | | | | | | | |
| CASALE | | | | | | | | 1 | | | | | | 1 | | 2 | |
| JUPENG BIO | | | | | | | | | | | | 1 | 2 | 1 | | | |
| LINDE | | | | | 2 | | 1 | | | | 1 | | | | | | |
| MEMBRANE TECHNOLOGY & RESEARCH | | | | | | | | 1 | 2 | | | | 1 | | | | |

**Figure 16.** Investment trend for key players. The figure uses two colors to represent high and low

numbers i.e., the more green represents high number and more red indicates a lower figure of investment trend each year.

## 9. Challenges Ahead in CCUS

There are several challenges that need to be addressed in order to advance the development and deployment of carbon capture, utilization, and storage (CCUS) technologies. These challenges include:

- Cost: CCUS technologies are currently expensive to implement, which limits their widespread adoption.
- Scale: CCUS technologies are still in the early stages of development and have not yet been deployed at a large scale. This makes it difficult to demonstrate their feasibility and effectiveness.
- Infrastructure: Implementing CCUS technologies often requires significant infrastructure investments, such as pipelines and storage facilities.
- Public acceptance: There is often resistance to CCUS technologies due to concerns about their potential impacts on the environment and communities.
- Legal and regulatory frameworks: CCUS technologies are regulated by a patchwork of laws and regulations that can vary from place to place. This can create uncertainty and make it difficult to advance the deployment of these technologies.
- Technological challenges: There are ongoing efforts to improve the efficiency and effectiveness of CCUS technologies, but there are still technological challenges that need to be addressed in order to make them more viable.
- Competition with other technologies: CCUS technologies face competition from other technologies that aim to reduce greenhouse gas emissions, such as renewable energy and energy efficiency measures.

Overall, addressing these challenges will require a combination of technological innovation, policy and regulatory support, and public engagement and education.

## 10. $CO_2$ Leakage Hazards

Carbon dioxide ($CO_2$) leakage is a potential hazard associated with the storage of $CO_2$ in underground geological formations. When $CO_2$ is injected into the ground for storage, there is a risk that it could leak out of the storage site and into the surrounding environment. This could pose a variety of risks, including:

- Environmental impacts: $CO_2$ is a greenhouse gas, and if it leaks into the atmosphere it can contribute to global warming and climate change.
- Human health impacts: If $CO_2$ leaks into the air, it can displace oxygen and cause respiratory problems in humans and other animals.
- Property damage: $CO_2$ leaks can cause damage to buildings and infrastructure, particularly if the $CO_2$ accumulates in enclosed spaces.

To mitigate the risk of $CO_2$ leakage, it is important to carefully select and monitor storage sites, use appropriate injection and monitoring technologies, and establish robust regulatory frameworks to oversee the storage of $CO_2$.

## 11. Regulatory and Policy Issues for CCUS

The adoption of carbon capture, utilization, and storage (CCUS) technologies has been slow due to a lack of government action on climate change, public skepticism, rising costs, and advances in other options such as renewables and shale gas. A comprehensive evaluation of various technologies or methods is necessary to reduce or eliminate $CO_2$ emissions. There is a need for comprehensive policies that can successfully reduce $CO_2$ while also saving energy and creating jobs in the 21st century economy. While research and development efforts are ongoing to address the technological challenges of effective CCUS implementation, a legislative framework is necessary to properly implement technologies and monitor their significant role in mitigating carbon emissions. CCUS regulations should

cover the regulatory treatment of $CO_2$ and other gases in the $CO_2$ stream, monitoring, verification, and remedial strategies to ensure that CCUS can effectively mitigate carbon emissions and provide a path to future hydrocarbon supplies.

## 12. The Future of CCUS

$CO_2$ could potentially be permanently stored in concrete through new processes, offering a decarbonization opportunity in building materials, such as precast structural concrete slabs and blocks. These materials could be made with cement types that, when cured in a $CO_2$-rich environment, create concrete containing around 25% $CO_2$ by weight. This could potentially be a significant way to reduce $CO_2$ emissions.

Carbon-neutral fuels for jets are a possibility through the use of $CO_2$ captured from industrial processes. By chemically reacting this $CO_2$ with hydrogen, synthetic fuels like gasoline, jet fuel, and diesel can be created. However, the production of sufficient amounts of hydrogen in a sustainable manner is crucial to the success of this approach.

The biomass-energy cycle can potentially be $CO_2$-neutral or even carbon-negative with the use of Bioenergy with Carbon Capture and Storage (BECCS). This process involves using sustainably harvested wood or other $CO_2$-rich biomass sources, such as algae, as a fuel source and capturing the resulting $CO_2$ emissions. The captured $CO_2$ can then be permanently stored, effectively removing it from the atmosphere and offsetting the carbon emissions from the fuel combustion process. This approach has the potential to be an effective method for mitigating climate change.

Carbon fiber, a super strong and lightweight material, is used in the production of a variety of products including airplane wings and wind turbine blades. The high price of carbon, a key component in the production of carbon fiber, has led manufacturers to search for a cheaper $CO_2$-derived alternative. The potential for using $CO_2$ in the production of carbon fiber on a large scale could significantly reduce costs and increase its use in building materials.

A new type of cement, made using a previously unused byproduct of bauxite mining as a raw material, has the potential to reduce $CO_2$ emissions by up to two thirds during production. This alternative cement has been shown to be just as stable as traditional Portland cement, making it a promising option for reducing the carbon footprint of the construction industry.

Carbon capture and storage (CCS) technologies can permanently store carbon dioxide ($CO_2$) in deep geological formations, creating negative emissions or carbon removal. Alternatively, $CO_2$ can be extracted from the atmosphere using direct air capture (DAC) technologies and used in various applications, such as food processing or the production of synthetic fuels through combination with hydrogen. These options offer the potential to significantly reduce $CO_2$ emissions and address the challenges of climate change.

## 13. Summary and Conclusions

- A review of Carbon capture utilization and storage potential in Baltic Sea Region (BSR) Countries have been presented.
- The patent landscape search has been conducted to identify patents related to Carbon capture and sequestration.
- The search has been conducted for Lithuania (LT) OR Latvia (LV) OR Estonia (EE) OR Finland (FI) OR Denmark (DK) OR Sweden (SE) OR Russia (RU) OR Poland (PL) OR Norway (NO) OR Iceland (IS) OR Spain (ES) and restricted with earlier Priority date from 2000 to 2020.
- Patents primarily focused on $CO_2$ storage, monitoring, utilization, and transport.
- Identified 497 patents as relevant to CCUS (capture, storage, utilization, and transport).
- Observed decreased priority filing activity from 2004 onwards. High filing rates observed in 2009 and 2011; 2009 has the greatest number of IP activity for CCUS.
- Russia (RU) has the highest number of publications followed by Poland (PL) and Norway (NO).

- A sudden surge in patent filing in Russia since 2007 onwards. Continued interest in patent filing is due to increased application-based industries coming up in the region.
- 2019 has the lowest number of granted patents and 2016 has the lowest number of filings.
- 50% of the relevant patents are for $CO_2$ capture.
- 85% of 497 relevant INPADOC families is alive. Russia (RU) has the greatest number of alive patents.
- Russia (RU) has the greatest number of patents/publications in $CO_2$ capture, storage and utilization/transport followed by Poland (PL) and Norway (NO).
- GE has the highest number of publications followed by Mitsubishi and Siemens.
- $CO_2$ capture is the most explored technology/CCUS type. Capture technology has been prominent from 2003 to 2013.
- Observed capture technology has the greatest number of patents/publications, general and post combustion being the top category followed by Oxyfuel.
- $CO_2$ separation/extraction and $CO_2$ from gas/flue gas is the most prominent in all the divisions.
- Y02C is the top CPC, which is related to CAPTURE, STORAGE, SEQUESTRATION OR DISPOSAL OF GREENHOUSE GASES and followed by B01D, which is related to Separation.

**Author Contributions:** All authors contributed equally in designing and drafting of the manuscript and in revisions. The first draft of the manuscript was written by all authors in a joint effort. All authors commented on previous versions of the manuscript. All authors read and approved the final manuscript.

**Funding:** Funding support from Nordic Energy Research is acknowledged for supporting 1st Author and covering publication cost for the paper. Work carried out by third author is supported by Lithuanian Research Council Funding for postdoctoral research fund Proposal registration No. P-PD-22-022-PATIKSLINTA.

**Data Availability Statement:** Not applicable.

**Acknowledgments:** Authors acknowledge the support of BASRECCS and its sponsor Nordic Council of Ministers. Authors would also like to acknowledge the support provided by TekIP knowledge consulting in analysis of patents and patent searches.

**Conflicts of Interest:** The authors declare no conflict of interest.

## Nomenclature

Country code

| | |
|---|---|
| RU | Russia |
| PL | Poland |
| NO | Norway |
| DK | Denmark |
| SE | Sweden |
| FI | Finland |
| ES | Spain |
| IS | Iceland |
| LT | Lithuania |
| LV | Latvia |
| EE | Estonia |
| EP | Europe |
| US | United States of America |
| CA | Canada |
| CN | China |
| Type | Overview |

| Publication Countries | Countries where the patents have been published |
| Patent Family | A patent family is the same invention disclosed by a common inventor(s) and patented in more than one country. |
| Earliest priority date | Earliest priority date is the first filing date within a family of patent applications |
| INPADOC | Stands for International Patent Documentation, is an international patent collection. The database is produced and maintained by the European Patent Office (EPO). It contains patent families and legal status information, and is updated weekly. |
| CPC | The Cooperative Patent Classification (CPC) is a patent classification system, which has been jointly developed by the European Patent Office (EPO) and the United States Patent and Trademark Office (USPTO). |

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
