# Peer review of "Exploring the Potential of Carbon Capture, Utilization, and Storage in Baltic Sea Region Countries: A Review of CCUS Patents from 2000 to 2022"

_processes, doi:10.3390/pr11020605_

Round 1
Reviewer 1 Report
The work is an important contribution and only suggest one last revision of the text and the graphics
Reviewer 2 Report
The MS title “Exploring the potential of Carbon Capture, Utilization, and Storage in Baltic Sea Region Countries: A Review of CCUS Pa-tents from 2000-2022” is a quite interesting topic. We must appreciate the authors to take initiative for such kind of studies especially for controlling carbon dioxide gas emissions. But, unfortunately, authors have not analyzed and discussed it properly. It is poorly and/or casually written. It just, looks like a report.
In this current condition, it can not be recommended for publication.
Reviewer 3 Report
The study “Exploring the potential of Carbon Capture, Utilization, and Storage in Baltic sea region Countries: A Review of CCUS Pa-tents from 2000-2022” is commendable and contemporary. Some improvements are needed. My suggestions are as follows;
1. It is better to choose the article type as a “review article” since a review has been presented rather than presenting new results.
2. Introduction section: International literature regarding the CCUS is missing. I would suggest the authors to also add the studies that were carried on a global scale. On page 1, there are no references, my suggestions regarding high-quality CCUS research are as follows;
a) A critical review on deployment planning and risk analysis of carbon capture, utilization, and storage (CCUS) toward carbon neutrality
b) Exploring the power of machine learning to predict carbon dioxide trapping efficiency in saline aquifers for carbon geological storage project
c) Application of robust intelligent schemes for accurate modelling interfacial tension of CO2 brine systems: implications for structural CO2 trapping
d) A multi-scale framework for CO2 capture, utilization, and sequestration: CCUS and CCU
e) Application of hybrid artificial intelligent models to predict deliverability of underground natural gas storage sites.
f) China's carbon capture, utilization and storage (CCUS) policy: A critical review
g) A review of optimization and decision-making models for the planning of CO2 capture, utilization and storage (CCUS) systems
h) Developing the efficiency-modeling framework to explore the potential of CO2 storage capacity of S3 reservoir, Tahe oilfield, China.
3. In the first section, introduction of research gap would be quite useful. Also, a separate literature conducted on the Baltic sea region Countries regarding should be gathered in a separate sub-section.
4. For section 3, it would be helpful to add a table regarding the CO2 usage technologies.
5. For figure 1, the black colored text is not readable.
6. For Figure 2 and 3, increase the font size.
7. For table 2 and 3, use the lowercase for the text.
8. The classification is presented. However, a sketch or a framework of the whole scenario is missing which should be included.
9. Future prospects are suggested, but pros and cons are missing. Limitations also needed to be included.
10. Some sections can be merged since 13 sections are too many. An overall improvement in the presentation is needed to increase the gist of study.
11. Conclusions should be improved. Key points in the form of bullet points are encouraged.
12. Add a nomenclature section since many abbreviations have been used.
13. Reference list needs to be updated. Also, follow MDPI style patter.
Round 2
Reviewer 3 Report
The authors have adequately addressed all my queries. The revised manuscript is an improved version and is acceptable for publication.